# Sedation with Sevoflurane versus Propofol in COVID-19 Patients with Acute Respiratory Distress Syndrome: Results from a Randomized Clinical Trial

**DOI:** 10.3390/jpm13060925

**Published:** 2023-05-31

**Authors:** Sara Martínez-Castro, Berta Monleón, Jaume Puig, Carolina Ferrer Gomez, Marta Quesada, David Pestaña, Alberto Balvis, Emilio Maseda, Alejandro Suárez de la Rica, Ana Monero Feijoo, Rafael Badenes

**Affiliations:** 1Department Anesthesiology, Surgical-Trauma Intensive Care and Pain Clinic, Hospital Clínic Universitari, University of Valencia, 46010 Valencia, Spain; sara.martinezcastro@hotmail.com (S.M.-C.);; 2Anesthesiology and Intensive Care Department, Consorcio Hospital General Universitario, 46014 Valencia, Spain; 3Anesthesiology and Intensive Care Department, Hospital Universitario Ramón y Cajal, 28034 Madrid, Spain; 4Surgical Critical Care Department, Hospital Universitario La Paz, 28046 Madrid, Spain; 5Anesthesiology and Surgical Critical Care Department, Hospital Universitario De La Princesa, 28006 Madrid, Spain

**Keywords:** sevoflurane, COVID-19, acute respiratory distress syndrome, deep sedation, critical care

## Abstract

Background: Acute respiratory distress syndrome (ARDS) related to COVID-19 (coronavirus disease 2019) led to intensive care units (ICUs) collapse. Amalgams of sedative agents (including volatile anesthetics) were used due to the clinical shortage of intravenous drugs (mainly propofol and midazolam). Methods: A multicenter, randomized 1:1, controlled clinical trial was designed to compare sedation using propofol and sevoflurane in patients with ARDS associated with COVID-19 infection in terms of oxygenation and mortality. Results: Data from a total of 17 patients (10 in the propofol arm and 7 in the sevoflurane arm) showed a trend toward PaO_2_/FiO_2_ improvement and the sevoflurane arm’s superiority in decreasing the likelihood of death (no statistical significance was found). Conclusions: Intravenous agents are the most-used sedative agents in Spain, even though volatile anesthetics, such as sevoflurane and isoflurane, have shown beneficial effects in many clinical conditions. Growing evidence demonstrates the safety and potential benefits of using volatile anesthetics in critical situations.

## 1. Introduction

Coronavirus disease (COVID-19) has become a significant worldwide challenge for health care providers. The wide variety of COVID-19 symptoms developed have ranged from mild headache or isolated cough to severe respiratory failure. Many patients affected by COVID-19 developed ARDS and required ICU admission for invasive mechanical ventilation (IMV). Oxygenation impairment and increased mortality rates have characterized this worldwide health crisis [1,2].

Impaired oxygenation in the ARDS context usually requires IMV support, and sedation is an integral part of therapy for this kind of patient. Titratable light-to-deep sedation may control neurological manifestations and help optimize ventilatory settings and endotracheal tube tolerance, although sometimes it is also necessary to use neuromuscular relaxants. Within the variety of sedative agents used in ICUs, there are two large groups classified according to administration route: intravenous and inhaled agents. Intravenous drugs represent routine clinical practice worldwide [3,4]: benzodiazepines (midazolam, lorazepam, and diazepam), propofol, and ketamine are commonly combined with opioids to achieve analgo-sedation. In contrast, volatile anesthetics (isoflurane and sevoflurane) do not have this privilege despite the potential beneficial effects demonstrated in animal models and patients suffering from ARDS [5,6,7].

Ideal sedation for ARDS-affected patients must be free of side effects, with rapid onset of action and a titratable dose–response relationship, with protective lung effects and with quick recovery time to facilitate weaning from mechanical ventilation. Nowadays, none of the available sedatives achieve 100% of these characteristics, but continuous improvements bring them closer to the ideal. Below, the main sedative families currently available are introduced.

Intravenous sedatives are widely used in ICUs worldwide and they are currently in the pole position, due to their well-known pharmacological properties, variety, efficacy, safety, and low price, as well as because clinicians are familiar with them. Periodically updated sedation guidelines base their recommendations on these drugs. However, sedation using intravenous drugs for long periods can be problematic due to potential accumulation (some active metabolites are organ-dependent for elimination), which can lead to unpredictable pharmacokinetics and pharmacodynamics, poor clearance, tolerance, withdrawal (with slow wake-up), delirium, or hemodynamic instability. Even if novel drugs have been commercialized, the main characters of this play are still propofol and midazolam, as dexmedetomidine is being increasingly used on a daily basis, but esketamine is not yet available in our country.

Inhaled agents (nitrous oxide, halothane, isoflurane, desflurane, and sevoflurane) can be used for induction and maintenance of general anesthesia in the operating room. Nitrous oxide and halothane have several serious side effects, and, nowadays, they are not commonly used for inhalation anesthesia. The popular drugs are isoflurane and desflurane because of rapid onset of action, high potency, no tachyphylaxis, and rapid offset due to clearance to lung exhalation. They require specific equipment and adequate ICU team training, but they have ideal pharmacological properties that allow efficient, well-tolerated, and easy titratable sedation. Moreover, there is no accumulation of toxic metabolites in critically ill patients as compared to benzodiazepines. The safety and efficacy of sevoflurane and isoflurane have been demonstrated in various studies [8,9,10] and continue to be under investigation [11], and technical equipment such as the anesthetic-conserving device (AnaConDa^®^; Sedana Medical, Sweden) has significantly simplified the application of volatile anesthetics in the ICU. In addition, these molecules may have clinical benefits that could be especially relevant to patients affected by ARDS, due to their immunomodulatory, anti-inflammatory, bronchodilator, and anti-thrombotic properties [12,13,14,15,16,17]. However, ICU members are not familiar with inhaled sedation, the reflection devices used with inhaled sedation (slightly) increase the instrumental dead space volume, and there is a potential risk of air pollution with their use. However, all these cons must be counteracted by the knowledge of these drugs, the optimal adjustment of ventilator settings and end tidal carbon dioxide monitoring, and the use of adequate scavenging systems for expired gases. Furthermore, the AnaConDa^®^ device has a carbon filter that absorbs and recycles more than 80% of the inhaled anesthetics.

Despite the fact that most anesthesiologists commonly use inhaled anesthetics in the operating room, it was not until the intravenous anesthetic stock shortage during the pandemic [6,18,19,20], and the non-benzodiazepine strategies preferred in ICU, that both volatile [21] and enteral agents [22] were taken into consideration. Sedation of ARDS COVID-19 patients became a great challenge. In addition, younger patients with higher sedative requirements and prone positioning, or those undergoing extracorporeal membrane oxygenation (ECMO) [23], justified the combination of both anesthetic groups (combined with opioid-based analgesics) to facilitate ventilator synchrony and tube tolerance. These facts gave our group the opportunity to design a clinical trial to assess the potential benefit of sedative agents on oxygenation in patients with ARDS due to COVID-19 infection. The initial idea was to recruit all consecutive patients admitted in participant critical care units, but emergency situations, patients’ social problems, lack of organization, and workers on sick leave made it difficult to perform. Our team achieved poor patient recruitment, but we decided to analyze these data so we can draw some conclusions to continue with our study.

## 2. Materials and Methods

### 2.1. Study Population, Setting and Data Collection

After review and approval by the ethics committee of the INCLIVA Health Research Institute, our team developed a multicenter, national, randomized 1:1, controlled, parallel, open study registered as NCT04359862, in which patients with ARDS due to COVID-19 were included in the first 24 h after diagnosis (Figure 1). The participant centers in this trial were four third-level hospitals in Spain which were referents during the COVID pandemic: Hospital Clínic Universitari of Valencia (Valencia), Consorcio Hospital General Universitario of Valencia (Valencia), Hospital Universitario Ramón y Cajal (Madrid), and Hospital Universitario La Paz (Madrid). The initial sample size took into consideration the high number of daily admissions in our units, but organizational problems made the recruitment rate much lower than expected. This trial was carried out during the year 2020.

Each patient who met the inclusion criteria and none of the exclusion criteria was proposed to participate in the study. When the informed consent was signed by the patient (or relatives when patient could not sign), randomization was performed to a treatment arm: sedation using sevoflurane (SEV) or propofol (PROP). In the PROP group, propofol was administered with volumetric pumps (Alaris GW and GP Plus), and sedation levels were evaluated using BIS^®^ technology (Medtronic Covidien, Spain). In the SEV group, sevoflurane was administered through an AnaConDa^®^ device coupled to a ContraFluran^®^ scavenging device, and CAM was measured with a SedLine^®^ monitor. Inclusion criteria were: age ≥ 18 years old, need for sedation, ARDS due to COVID-19, and accepted informed consent by the patient or a relative. Exclusion criteria included intracranial hypertension, allergy to any sedative agent, tidal volume < 250 mL, previous malignant hyperthermia or risk of developing malignant hyperthermia, hepatic failure, neutropenia, pregnancy, or chemotherapy in the previous month. All randomized patients received remifentanil as an analgesic and cisatracurium as a neuromuscular relaxant. A lung-protective ventilation strategy was carried out: V_T_ 6 mL/kg, PEEP > 5 cmH_2_O, Plateau pressure < 30 cmH_2_O, respiratory rate < 35 rpm, and I:E ratio ≤ 1:2 [24]. This study protocol is publicly available for verification [25].

Patients were anonymized using a numerical code for data collection. Variables recorded included: anthropometric values, respiratory and hemodynamic parameters, and blood and bronchoalveolar fluid samples at day of randomization (D0), at 24 h (D1), and at 48 h (D2) (see Table 1). Thirty days after randomization (D30), the following data were collected: duration and control of mechanical ventilation (MV), ventilator-free days (VFD), length of ICU stay, and mortality at D2 and D30.

### 2.2. Hypothesis and Objectives

The hypothesis of this clinical trial was that sedation using sevoflurane in patients with ARDS associated with COVID-19 infection improves oxygenation. The primary objective included the evaluation of oxygenation in randomized patients during the first 48 h, measured via PaO_2_/FiO_2_. The secondary objective included assessment of mortality rates at D30.

### 2.3. Statistical Analysis

All statistical comparisons were based on the intention-to-treat principle. Sample size was calculated using results of previous works [7,14]. The Student’s *t*-test and the Mann–Whitney test were used as appropriate. Chi-square and Fisher’s tests were used for categorical variables. Differences in PaO_2_/FiO_2_ for D1 and D2 between the two treatment arms were determined using mixed regression analysis for repeated measures. This method also included an ANCOVA-type design; baseline values of the PaO_2_/FiO_2_ variable were added to the regression model as covariates, interacting with the treatment variable. The analysis was also adjusted for the potential of autocorrelation between repeated measures in the same subject and for the nesting effect (due to the study center) through “random intercept” effects. The Kaplan–Meier method, Cox regression, and restricted median survival time (RMST) were used to compare D30 survival rates between the two study groups. Statistical analysis was performed using Stata version 16.1 (StataCorp. 2021. Stata Statistical Software: Release 16. College Station, TX, USA: StataCorp LP).

## 3. Results

### 3.1. Primary Results

Analyses were based on a total of 17 patients: 10 patients in the PROP arm and 7 patients in the SEV arm. In Valencia, HCUV recruited seven patients and Consorcio Hospital General Universitario recruited three. In Madrid, Ramón y Cajal Hospital recruited one patient and La Paz Hospital recruited six. For each patient, multiple variables were collected on D1 and D2, and mortality was noted on D30. Regarding the patients’ baseline characteristics, no statistically significant differences were found in any of the variables listed in Table 2. No statistically significant difference was found regarding several co-morbidities (stroke, arterial hypertension, diabetes mellitus, dyslipemia, smoking habit, chronic kidney injury (CKI), and previous corticosteroid therapy). The calculated severity index (via SAPS-II [26] and LIS [27]) did not show differences between groups.

Patients in the PROP arm spent an average 16.4 days in the ICU, and patients in the SEV arm spent an average 20.6 in the ICU (*p* = 0.563); the average days under IMV were 13 (PROP) vs. 14.6 (SEV), with the average IMV-free days in the ICU being 5.1 vs. 5.2, respectively. The mean number of days from randomization to death was 28 vs. 30, respectively (*p* = 0.495). Regarding ventilatory settings, no significant differences were found at baseline (Table 3) or at D1 (Table 4). At D2 (Table 5), statistically significant differences among groups were found; respiratory acidosis developed in the SEV group, probably related to differences in end-expiratory lung volumes.

### 3.2. Effect on PaO_2_/FiO_2_

Figure 2 illustrates the changes in PaO_2_/FiO_2_ between the two treatment arms at the baseline (before treatment), and at D1 and D2 (post-treatment), as previously shown in Table 2, Table 3, Table 4 and Table 5. As shown in Figure 2, there were differences between the groups before randomization, with higher PaO_2_/FiO_2_ values in the SEV group (*p* = 0.246). Differences persisted at D1 and minimized at D2.

Figure 3 presents the core of the analysis: the effect of the randomized treatment on PaO_2_/FiO_2_ (at D1 and D2, post-randomization). It was adjusted by the baseline value of PaO_2_/FiO_2_ according to the ANCOVA design. Using mixed regression analysis in the context of ANCOVA, the results showed an improvement in the PaO_2_/FiO_2_ ratio in the SEV arm at D1 and D2; however, results were statistically significant only at D1** (Figure 3).

### 3.3. Effect over 30-Day Mortality

Kaplan–Meier analysis (Figure 4) showed an intersection of survival curves along the trace, which made it difficult to interpret and find the meaning of the Log-rank test (*p* = 0.584). Therefore, we tested a time-dependent effect for the PaO_2_/FiO_2_–mortality relationship. On average, patients treated using sevoflurane survived 1.66 days longer than those treated using propofol when patients were followed up at D30 (Figure 5). Unfortunately, this difference did not achieve statistical significance (95% CIs = −11.00 to 14.33). Figure 6 shows a time-dependent effect with a higher mortality risk at the beginning of the study for the SEV arm, but this risk decreased to become protective in the following period. Unfortunately, confidence intervals reached the 1-line on the Y-edge, so the analysis did not achieve statistical significance.

The prognostic effect of PaO_2_/FiO_2_ differences between D1 and D2 compared to the basal values for the SEV and PROP arms was tested using sensitivity analysis. Figure 7 shows Cox regression analysis results for D30 mortality, in which no statistical significance was found for changes in PaO_2_/FiO_2_. This showed that the effect of treatment on mortality did not depend on changes in PaO_2_/FiO_2_ measured at D1 and D2 compared to basal values.

## 4. Discussion

### 4.1. Limitations

The main limitation of this clinical trial was its sample size. The health emergency situation in which this project was carried out led to the loss of recruitment and clinical data (due to work overload and difficulties in obtaining informed consent). In addition, assessment of oxygenation, levels of inflammatory mediators, and mortality in patients admitted to the ICU for ARDS associated with COVID-19 were probably influenced by unknown factors due to the incipient development of infection by this virus. Moreover, the study’s initial protocol included cytokine level measurements; however, these could not be assessed due to laboratory issues (overwhelmed by COVID-19 tests) and storage limitations.

### 4.2. Generalizability and Interpretation

Intravenous agents (mainly propofol and benzodiazepines (3)) have been the most-employed deep sedation drugs in ICUs worldwide. That is surprising, as the use of benzodiazepines has been associated with decreased ventilator-free days, increased risk of delirium, and worse long-term outcomes [28,29], so non-benzodiazepine strategies should be preferred for ICU sedation.

The COVID-19 pandemic emptied all hospital sedative stocks in just a few months. The main reason for this was that ICU capacities were overrun with an increase in invasively mechanically ventilated ARDS patients who required deep sedation combined with muscle relaxation to achieve a depth of sedation sufficient to avoid patient–ventilator dyssynchrony. At the beginning of the pandemic, some groups emphasized that affected patients required high sedative doses; although some possible underlying reasons for this were unknown, rational reasons included that patients were younger, without co-morbidities, and most of them needed to toggle prone position.

This situation necessitated opening the therapeutic arsenal to other options, such as volatile anesthetics and multimodal sedative approaches, to avoid adverse effects related to propofol (such as hyperlipidemia or propofol infusion syndrome (PRIS)) or the abuse of neuromuscular relaxants or analgesics (mainly opioids). Inhaled sedation (using isoflurane or sevoflurane) had already demonstrated faster and improved recovery after prolonged sedation [30,31], minimized delirium, an analgesic sparing effect [32,33,34], decreased pulmonary inflammation [14,35], improved oxygenation in patients with ARDS [5], and decreased mortality in long-term ventilated patients [36]. Sevoflurane has been used in many ICUs since AnaConDa© was designed and the accuracy of the pharmacokinetic model was published [37]. Some guidelines included its use in critically ill patients with ARDS for a moderate-to-deep sedation more than ten years ago [38,39]. Still, many Spanish ICUs did not switch to inhaled sedation until they ran out of propofol; the main reasons provided were that staff were unfamiliar with the volatile agent or its specific device (AnaConDa©), even though it was widely used for intraoperative anesthetic maintenance.

The primary role that propofol and midazolam have in ICUs increases the complexity of changing the routine from intravenous sedation in order to assess the spectrum of sedative agents that we currently use. Profiling the potential benefit of a hypnotic agent over oxygenation is complicated given the interference of multiple variables that probably act as confounding factors in a critically ill patient. However, it is important to keep in mind that the profile of these drugs (that act at so many levels) must be recognized in order to adapt decisions based on the patient being treated [40]. In addition to its function as sedative agent, sevoflurane has many intrinsic characteristics with potential therapeutic benefits that could be especially relevant to ICU patients: it is an easy-to-titrate drug with shorter wake-up times, it has enhanced effect over analgesics (decreased use of opioids) and neuromuscular relaxants, and it has fewer vasopressor requirements compared to midazolam and propofol. All of these potential benefits should be taken into consideration [34,41].

Our study had inherent design limitations that made it difficult to draw categorical conclusions. However, this study highlights the feasibility of using sevoflurane as a primary sedative agent in ARDS patients. Both PROP and SEV treatment arms were comparable regarding patients’ characteristics, co-morbidities, and ventilatory settings. Regarding the PaO_2_/FiO_2_ ratio, it tended to improve in the SEV group both at D1 and D2; however, results were statistically significant only for D1. The SEV arm experienced longer ICU stays and longer days under MV; however, the number of days from randomization to death was longer in the SEV group. Moreover, patients on sevoflurane survived longer than those on propofol when patients were followed up at 30 days, even if results were not statistically significant. The reason for not achieving significance could be both low sample size and days under inhaled sedation; a retrospective study in surgical ICU ventilated patients (*n* = 128) with inhaled drugs used for more than 96 h demonstrated more ventilator-free days at day 60, more hospital-free days at 6 months, and decreased mortality compared with patients under intravenous sedation receiving midazolam or propofol [36]. Our results are consistent with previous analysis [30,42,43,44,45], which found no differences between inhaled and intravenous sedation in deaths or length of ICU stay. An international retrospective study including 10 ICUs published in 2022 [46] found no association between inhaled sedation in COVID-19 patients and the number of ventilator-free days through to day 28; this suggests that the effect of treatment on mortality probably does not depend on the resulting changes in the PaO_2_/FiO_2_ ratio.

Regarding oxygenation, studies performed in mice, rat, and pig models of ARDS found that inhaled agents reduced alveolar and systemic levels of pro-inflammatory cytokines [7,14,47,48,49], improved arterial oxygenation, and decreased lung alveolar oedema [7,50]. Results of this work agree with previous publications in which the potential benefit of sevoflurane over oxygenation was observed [15,40]; however, more studies recruiting a higher number of patients are needed to support the use of inhaled agents. Few studies registered on the ICH GCP website include objectives regarding the study of sevoflurane in patients with moderate to severe ARDS diagnoses. Even so, volatile agents (sevoflurane and isoflurane) are used as alternatives to intravenous sedation in ICUs by an increasing number of physicians [51] as monotherapy or as part of a combined therapy [52]. There are some detractors because of volatile agents’ potential adverse events [53], but there is relevant literature that supports their feasibility and safety of use, without the risk of tolerance or effects on renal or liver function [30,34,45,54,55].

In our unit’s experience, managing one more drugs, whether or not they are superior to another agent, allows us to provide alternatives that can be beneficial to our various patients. Therefore, while waiting for new studies, the use of inhaled sedation with sevoflurane as the first line in patients affected by ARDS must be considered, and not only as second or third-line treatment, as recommended recently [28].

## 5. Conclusions

This study has demonstrated that sedation using sevoflurane improved oxygenation and increased survival times in patients affected by ARDS due to COVID-19 infection compared to propofol. Hence, in patients with ARDS who require sedation, sevoflurane is a safe and effective option that, in addition to its main purpose, has a beneficial effect on oxygenation and survival. Therefore, it could be considered as a first-choice strategy for this patient profile.

## Figures and Tables

**Figure 1 jpm-13-00925-f001:**
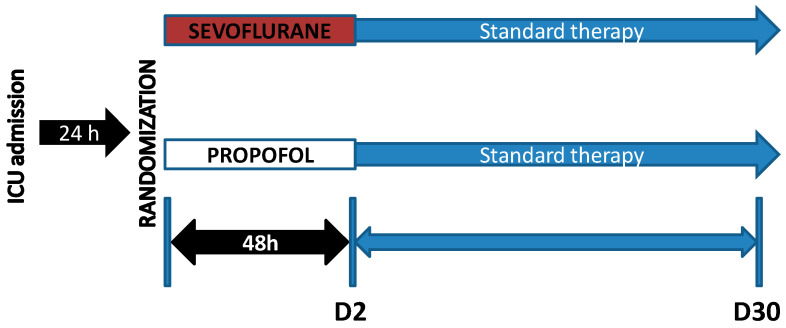
Protocol diagram with both intervention arms in the clinical trial. D2: day 2, D30: day 30.

**Figure 2 jpm-13-00925-f002:**
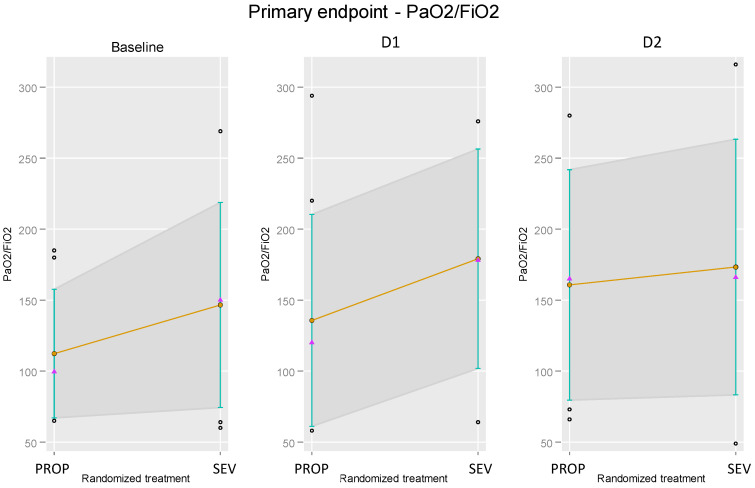
PaO_2_/FiO_2_ changes in the two treatment arms at the baseline (before randomization), and at D1 and D2 (post-randomization).

**Figure 3 jpm-13-00925-f003:**
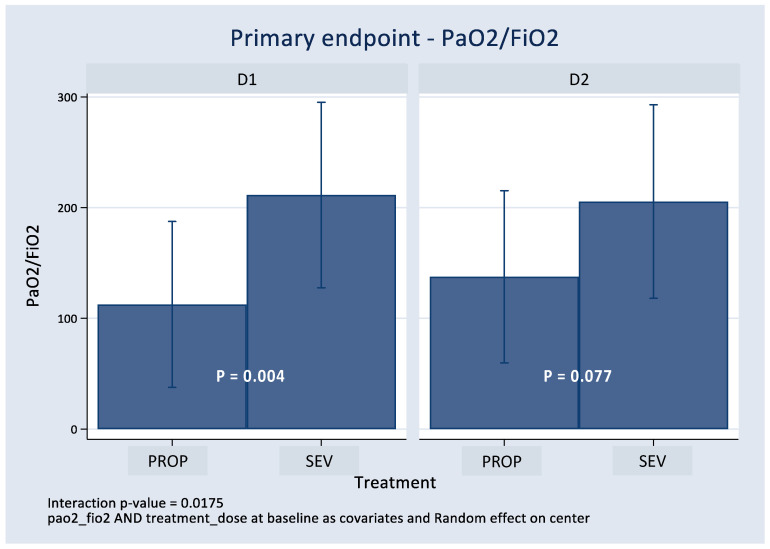
Effect of the randomized treatment on PaO_2_/FiO_2_ (at D1 and D2 post-randomization).

**Figure 4 jpm-13-00925-f004:**
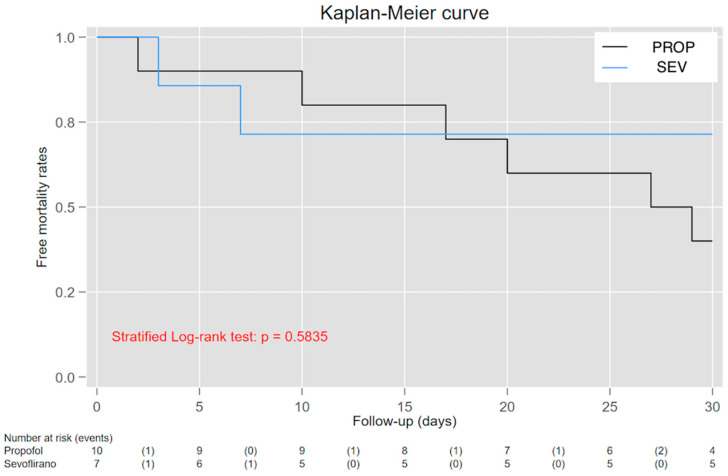
Survival curves for both treatment arms and their relationships with the D30 mortality endpoint using the Kaplan–Meier method.

**Figure 5 jpm-13-00925-f005:**
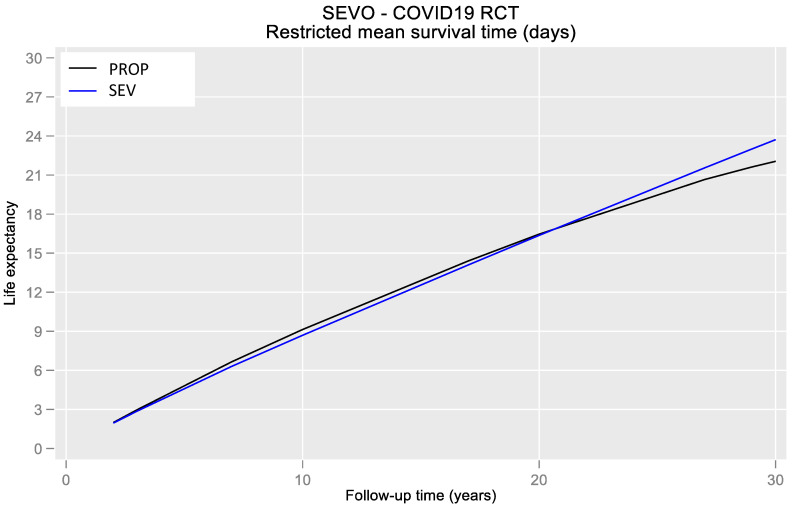
Restricted mean survival time (RMST) in days of the two treatment arms.

**Figure 6 jpm-13-00925-f006:**
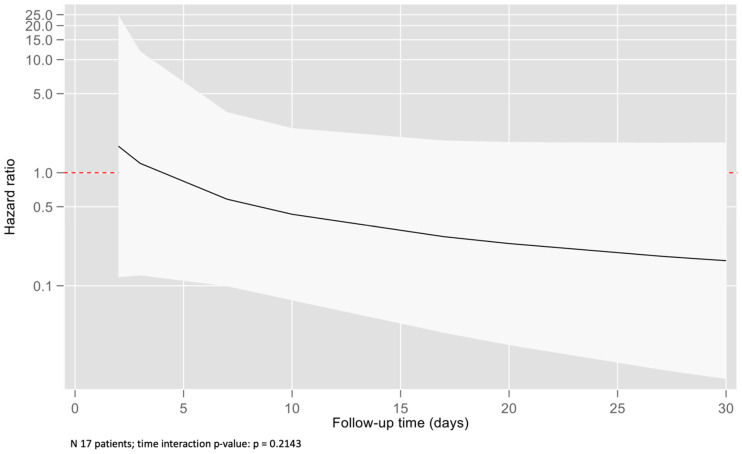
Hazard ratio curve for D30 mortality between sevoflurane and propofol along follow-up period.

**Figure 7 jpm-13-00925-f007:**
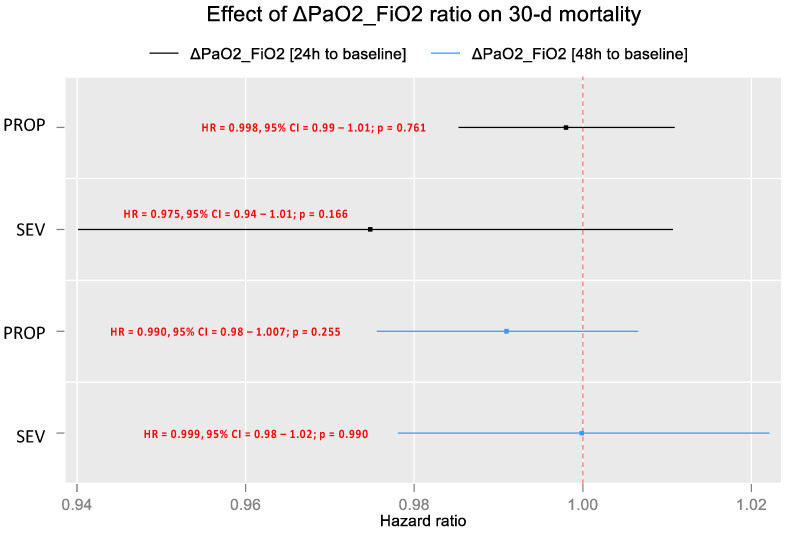
Predictive model using Cox regression of D30 mortality for the SEV and PROP arms, regarding PaO_2_/FiO_2_ changes.

**Table 1 jpm-13-00925-t001:** Visits and evaluations timetable.

Visits	Selection	Randomization	Visit 1	Visit 2	Final Visit
Time	−24 h	D0	D1	D2	D30
Procedures and testing					
Informed consent	X				
Inclusion/exclusion criteria	X				
Medical records	X				
Anthropometric data	X				
Clinical data	X				
Respiratory values	X				
Hemodynamic values	X				
Blood sample		X			
Respiratory sample		X			
Treatment					
Randomization		X			
Sedative agent administration		X	X	X	
Other treatments		X	X	X	X
Adverse events			X	X	X

**Table 2 jpm-13-00925-t002:** Patient characteristics. BMI: body mass index, SD: standard deviation.

Patient Characteristics	PROP	SEV	Total	*p* Value	Missings
Age, years, mean (sd)	62.9 (12.1)	64.4 (14.6)	63.5 (12.7)	0.816	0/17 (0.00)
Gender (female), *n* (%)	4 (40.0)	4 (57.1)	8 (47.1)	0.486	0/17 (0.00)
Height, m, mean (sd)	1.66 (0.09)	1.68 (0.06)	1.67 (0.08)	0.613	1/17 (5.88)
Weight, kg, mean (sd)	87.8 (31.4)	85.1 (22.3)	86.6 (27.0)	0.854	1/17 (5.88)
BMI, (kg/m^2^), mean (sd)	31.1 (7.4)	29.8 (6.0)	30.5 (6.6)	0.709	1/17 (5.88)

**Table 3 jpm-13-00925-t003:** Baseline ventilatory settings.

Baseline Ventilatory Settings, Mean (SD)	PROP	SEV	Total	*p* Value	Missings
End-expiratory lung volume, mL/kg	6.2 (1.0)	7.0 (1.8)	6.5 (1.4)	0.293	2/17 (11.76)
PEEP, cmH_2_O	8.6 (2.9)	9.7 (1.5)	9.1 (2.4)	0.365	0/17 (0.00)
Inspiratory pressure, cmH_2_O	15.6 (10.4)	22.0 (2.2)	18.4 (8.4)	0.170	3/17 (17.65)
Lung compliance, L/cmH_2_O	41.5 (24.4)	41.2 (8.5)	41.4 (18.6)	0.975	3/17 (17.65)
Airway resistance, cmH_2_O/L/s	19.5 (20.3)	15.0 (3.8)	17.6 (15.3)	0.605	3/17 (17.65)
FiO_2_ (%)	83.0 (17.0)	79.3 (19.7)	81.5 (17.7)	0.684	0/17 (0.00)
Arterial pH	7.35 (0.10)	7.37 (0.14)	7.36 (0.12)	0.759	0/17 (0.00)
Respiratory rate, breaths per minute	21.9 (4.1)	18.2 (3.9)	20.4 (4.3)	0.104	2/17 (11.76)
PaCO_2_, mmHg	48.5 (15.2)	45.4 (14.2)	47.2 (14.4)	0.680	0/17 (0.00)
PaO_2_/FiO_2_	112.3 (45.3)	146.6 (72.2)	126.4 (58.4)	0.246	0/17 (0.00)
Heart rate, beats per minute	64.1 (33.8)	81.9 (15.7)	71.4 (28.6)	0.217	0/17 (0.00)
Mean arterial pressure, mmHg	82.0 (12.6)	79.0 (13.5)	80.8 (12.7)	0.646	0/17 (0.00)
BIS	44.0 (9.3)	41.0 (5.2)	42.5 (7.4)	0.470	3/17 (17.65)

**Table 4 jpm-13-00925-t004:** Ventilatory settings on D1.

Ventilatory Settings on D1, Mean (SD)	PROP	SEV	Total	*p* Value	Missings
End-expiratory lung volume, mL/kg	11.9 (18.7)	6.1 (0.4)	9.5 (14.3)	0.432	0/17 (0.00)
PEEP, cmH_2_O	9.9 (3.4)	10.1 (1.7)	10.0 (2.7)	0.864	0/17 (0.00)
Inspiratory pressure, cmH_2_O	22.9 (5.5)	21.0 (4.1)	22.1 (4.9)	0.501	3/17 (17.65)
Lung compliance, L/cmH_2_O	33.8 (14.8)	44.3 (14.7)	38.7 (15.2)	0.191	2/17 (11.76)
Airway resistance, cmH_2_O/L/s	10.4 (8.4)	14.3 (5.5)	12.1 (7.3)	0.338	3/17 (17.65)
FiO_2_ (%)	81.5 (21.1)	69.9 (16.3)	76.7 (19.6)	0.240	0/17 (0.00)
Arterial pH	7.39 (0.07)	7.35 (0.14)	7.37 (0.10)	0.381	0/17 (0.00)
Respiratory rate, breaths per minute	20.5 (5.2)	17.0 (3.8)	19.1 (4.9)	0.151	0/17 (0.00)
PaCO_2_, mmHg	48.1 (12.5)	52.0 (15.2)	49.7 (13.4)	0.570	0/17 (0.00)
PaO_2_/FiO_2_	135.7 (74.6)	179.1 (77.3)	153.6 (76.5)	0.262	0/17 (0.00)
Heart rate, beats per minute	76.4 (18.9)	66.0 (30.1)	72.1 (23.8)	0.393	0/17 (0.00)
Mean arterial pressure, mmHg	73.2 (9.2)	78.3 (10.8)	75.3 (9.9)	0.311	0/17 (0.00)
BIS	44.0 (9.3)	41.0 (5.2)	42.5 (7.4)	0.470	3/17 (17.65)

**Table 5 jpm-13-00925-t005:** Ventilatory settings on D2.

Ventilatory Settings on D2, Mean (SD)	PROP	SEV	Total	*p* Value	Missings
End-expiratory lung volume, mL/kg	11.8 (18.7)	7.3 (2.6)	9.9 (14.3)	0.540	0/17 (0.00)
PEEP, cmH_2_O	10.1 (2.6)	9.9 (1.7)	10.0 (2.2)	0.832	0/17 (0.00)
Inspiratory pressure, cmH_2_O	21.1 (5.1)	24.2 (4.4)	22.4 (4.9)	0.268	3/17 (17.65)
Lung compliance, L/cmH_2_O	34.0 (15.8)	37.4 (13.9)	35.7 (14.4)	0.674	3/17 (17.65)
Airway resistance, cmH_2_O/L/s	12.9 (11.2)	14.2 (3.4)	13.4 (8.5)	0.792	3/17 (17.65)
FiO_2_ (%)	71.5 (15.3)	67.1 (16.8)	69.7 (15.6)	0.587	0/17 (0.00)
Arterial pH	7.42 (0.04)	7.31 (0.11)	7.38 (0.09)	0.010 **	0/17 (0.00)
Respiratory rate, breaths per minute	19.7 (3.6)	18.7 (4.9)	19.3 (4.0)	0.635	0/17 (0.00)
PaCO_2_, mmHg	44.5 (7.8)	59.1 (15.5)	50.5 (13.4)	0.021 **	0/17 (0.00)
PaO_2_/FiO_2_	160.7 (81.2)	173.3 (90.0)	165.9 (82.4)	0.767	0/17 (0.00)
Heart rate, beats per minute	73.6 (20.5)	80.7 (15.4)	76.5 (18.4)	0.451	0/17 (0.00)
Mean arterial pressure, mmHg	78.2 (13.6)	71.3 (7.9)	75.4 (11.8)	0.247	0/17 (0.00)
BIS	40.1 (10.0)	44.0 (19.1)	41.9 (14.5)	0.624	2/17 (11.76)

** Statistically significant result.

## Data Availability

Data supporting reported results are available from the corresponding author at any time.

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
