# Peer review of "Sedation with Sevoflurane versus Propofol in COVID-19 Patients with Acute Respiratory Distress Syndrome: Results from a Randomized Clinical Trial"

_jpm, 2023, doi:10.3390/jpm13060925_

Round 1

Reviewer 1 Report

This manuscript tries to show the superiority of sevoflurane over propofol in sedation of COVID-19 patients with ARDS.

I'm curious about some of the methods. (Major revision)

1. How did you get consents from the patients? Informed consent is one of the most important steps in terms of ethics.

2.  How did you apply propofol and sevoflurane? Dosage? Which machine you used to administer propofol and sevoflurane?

3. How did you calculate the sample size? It seems to small. Please specify the method you used for sample size calculation.

Results

1. Please use * to indicate significant results.

2. Figures: Use D1, D2 instead of 1-day, 2-day or visita dia1.

Author Response

Thank you for your first recommendation: we had an extensive English editing performed.

Regarding the introduction, we have modified it to provide more information regarding background and we added some more relevant references.

Regarding methods, design should be improved and for following study in this area we will improve this part for sure.

Regarding conclusions, we acknowledged that limitations regarding number of patients and analysis make it difficult to address clear conclusions. Hopefully, next part of our investigation will let us have stronger ones.

This manuscript tries to show the superiority of sevoflurane over propofol in sedation of COVID-19 patients with ARDS.

I'm curious about some of the methods. (Major revision)

  1. How did you get consents from the patients? Informed consent is one of the most important steps in terms of ethics.

After diagnosis of ARDS due to COVID19, in patients who were still awake, we asked for permission to participate in the study. But most of them came from Emergency service already sedated and intubated or with neurological symptoms due to hypoxia. In those cases, informed consent was signed by relatives (informed by phone call and signed at our ICU department). 

  1. How did you apply propofol and sevoflurane? Dosage? Which machine you used to administer propofol and sevoflurane?

Sedation/Anesthetic plane for all patients was monitored through Bispectral Index (BIS) Monitoring System. Moreover, patients´ CAM was measured using Massimo monitor for those under sevoflurane sedation. With both measurements, clinicians were adjusting anesthetic dosage.

Sedation was applied by intravenous infusion in case of propofol or inhaled for sevoflurane. Propofol was administered with volumetric pumps Alaris GW and GP Plus, as we do not have target-controlled-infusion pumps. Sevoflurane was administered through AnaConDa device. We also had ContraFluran as scavenging device.

  1. How did you calculate the sample size? It seems to small. Please specify the method you used for sample size calculation.

We could not calculate sample size as in that sanitary crisis we decided to recruit all potential patients but it was very difficult because of work overload. That is why only a small sample size was recruited.

Results

  1. Please use * to indicate significant results.

Done.

  1. Figures: Use D1, D2 instead of 1-day, 2-day or visita dia1.

Changed. Thank you for the valuable recommendations.

Reviewer 2 Report

Grammar issues

i.e. Sedation may cope (?) with neurological manifestations

i.e. endotraqueal

i.e. and deserve an honorary (i think it is more than honorary) position

Check all the grammar issues in the draft and the use of the adjective in a proper way

The main hypothesis is that Sevoflurane may improve oxygenation compared to propofol sedation: I think it might be interesting to explain in the discussion w the pharmacological rationale behind this hypothesis. I see a little redundancy when you explain both in the introduction and in the discussion that Covid gave us the possibility to explore Sevo sedation due to supply reduction

"Sedation with sevoflurane in patients affected by ARDS due to COVID-19 infection has demonstrated improved oxygenation and increased survival times compared to propofol" that is a strong statement for only 17 patients. I see your study as an explorative trial...it might improve oxygenation and again maybe concentrate more on the reason why it can be...

PS: do you have any information, beyond comorbidities, on the Covid presentation in the patients included in the study? First positive test, clinical course before ICU admission, and ET intubation? CT scan? I think these factors may influence oxygenation responses with both Sevo or prof sedation

Author Response

Thank you for your valuable recommendations that substantially improved our work.

Grammar issues

i.e. Sedation may cope (?) with neurological manifestations.

Changed.

i.e. endotraqueal

Changed.

i.e. and deserve an honorary (i think it is more than honorary) position

Changed, thank you very much.

Check all the grammar issues in the draft and the use of the adjective in a proper way

Our text has been under extensive English review to improve it.

The main hypothesis is that Sevoflurane may improve oxygenation compared to propofol sedation: I think it might be interesting to explain in the discussion w the pharmacological rationale behind this hypothesis. I see a little redundancy when you explain both in the introduction and in the discussion that Covid gave us the possibility to explore Sevo sedation due to supply reduction

Thank you for the recommendation. We mentioned it in both introduction and discussion because it was the main factor for sedation practice changes, but we performed some changes for it not to be so redundant. Thank you for your recommendation.

"Sedation with sevoflurane in patients affected by ARDS due to COVID-19 infection has demonstrated improved oxygenation and increased survival times compared to propofol" that is a strong statement for only 17 patients. I see your study as an explorative trial...it might improve oxygenation and again maybe concentrate more on the reason why it can be...

 We totally agree with you but the main objective of this study was to study oxygenation and mortality in inhaled vs intravenous sedation. In most of the references used, pharmacological reasons for improvement with sevoflurane are well-mentioned. In spite of that, we have added some reasons that could explain benefits of sevoflurane.

PS: do you have any information, beyond comorbidities, on the Covid presentation in the patients included in the study? First positive test, clinical course before ICU admission, and ET intubation? CT scan? I think these factors may influence oxygenation responses with both Sevo or prof sedation

We have all these data as database is wider than reflected in the study. In spite of that, our work did not have the objective of studying the relevance of those parameters. We take your comments into consideration for following part of the study as it can be really important.

Round 2

Reviewer 1 Report

I'm very sorry to tell you that I reject this manuscript. You did give answers to the methodological points I requested previously (eg informed consents, study drug dosage, method of use, etc), but none of them were revised in the manuscript. Even the point about the figure is not completely corrected.

Author Response

We are very sorry to hear that you have rejected our manuscript. It has been a huge work in very difficult conditions and all the team did their best. Also, we have performed some new and valuable corrections in the manuscript so we ask you to take them into consideration one more time. 

Regarding

  • Material and methods: initial sample size took into consideration the high number of daily admissions in our units, but organizational problems made the recruitment rate much lower than expected, so we have reflected it in the main text. Also, we have already mentioned the participant hospitals.
  • Results: even if the differences in lung volume are not statistically different, pH has clear differences. Your approach is correct, and we have reflected it in the main text. Despite that, to assume differences in survival related to those results, we should perform a wider study with a bigger n.
  • Limitations: we have reflected them in the main text. Future research in the same line will let us reproduce this work to draw more powerful conclusions.

Hoping you will take our changes into consideration.

Kind regards,

Reviewer 2 Report

I'm satisfied with the atuthor's responses to my questions. 

Author Response

Thank you very much for your previous valuable recommendations.

Kind regards,